# EXACT UNLEARNING OF FINETUNING DATA VIA MODEL MERGING AT SCALE

## ABSTRACT

Approximate unlearning has gained popularity as an approach to efficiently update a model so it (roughly) behaves as if it was not trained on a subset of data. However, approximate unlearning methods have been shown to be quite brittle in practice. In fact, such approaches can easily be attacked to reveal supposedly unlearned information. To address this issue, we instead propose a *model merging* approach, `ClAMU`, which produces combined models that can support both *efficient and exact* deletion of unlearning data. In addition to leveraging techniques from model merging and localization, `ClAMU` relies on two key innovations. First, we cluster tasks together and serve per-cluster models, balancing the tradeoff between the utility of local models versus the storage cost of a global model. Second, unlike existing localization methods which compress local models into masks, we propose directly optimizing local (or cluster-level) masks, which greatly improves utility. Relative to model merging and localization baselines, `ClAMU` serves models with up to 20% improved accuracy while reducing storage costs by up to 75%.

## 1 INTRODUCTION

Many modern applications of machine learning require finetuning a pretrained model on a collection of data. Once a model has been finetuned, it may be necessary to *unlearn* a subset of data and produce a model identical to one trained as if the data were never present. This is because finetuning can introduce risks such as learning harmful model behavior or exposing private information (Carlini et al., 2023; More et al., 2024; Su et al., 2024; Ahmadian et al., 2024). Moreover, data privacy regulations such as GDPR and CCPA state that consumers have a "right to be forgotten". For ML, this not only requires that data controllers delete user data in accordance with removal requests, but also retrain any models trained on the data. To address these concerns, there has been significant recent interest in methods for machine unlearning that can efficiently remove the influence of data from a model (Cao & Yang, 2015; Ginart et al., 2019; Bourtoule et al., 2021; Tarun et al., 2023).

In this work, we propose using *model merging* for both exact and efficient unlearning. Given a dataset split over several tasks (i.e. potential forget sets), we first finetune the pretrained model separately on each task to obtain a set of *local* models. Then, we *merge* the local models' weights to produce a single *global* model. Finally, we discard the local models, since storing all local models is prohibitive. To unlearn a particular task, we *deterministically* retrain the local model for that task and subtract it from the global model. When merging is a simple additive operation (e.g. averaging), subtracing the model provides exact unlearning. This approach has several efficiency benefits, namely (1) cheap execution of merging, (2) cheap storage cost of a single global model, and (3) cheap unlearning via training on the forget set rather than the retain set.

While model merging provides efficient and exact unlearning by design, it faces a critical issue of *task scaling*. To support a large number of unlearning requests (tasks), we must merge many models—one from each task. While prior work has only considered merging a small number of models (e.g., 2-30), we attempt to merge a large number of models (up to 500). We find that depending on the task data, merging quality can vary significantly between improving or degrading local performance.

When merging alone results in a poor global model, *localization* is a promising addition which learns masks from combining the global and local models — applying a mask to the global model recovers task-specific utility (Wang et al., 2024; Huang et al., 2024). Like merging, localization

has only been tested at small scales, and we show that localization also has limited utility at larger scales. Furthermore, localization introduces non-trivial costs which make unlearning less efficient than desired. Specifically, while local masks reduce storage by $\sim 32\times$ over naively storing all the local models, this cost still scales linearly with the number of tasks. Furthermore, in order to provide exact unlearning, all the local masks must be reconstructed after unlearning.

To address these challenges, we propose `ClAMU`, a framework that uses **CL**ustering, **A**veraging, and **M**asking for **U**nlearning. `ClAMU` makes two key modifications to adapt merging and localization for large scale unlearning. First, we *cluster* tasks together and learn *cluster-level* masks rather than task-level masks. Second, we improve localization by optimizing masks on training data. Overall, these techniques allow `ClAMU` to outperform similar baselines in all dimensions: utility, storage, and unlearning cost. Our contributions are as follows:

1. We merge up to 500 models finetuned on sharded vision and language datasets. We identify settings where the merged model is within (both above and below) 15% accuracy of the local models, making it a suitable choice for exact and efficient unlearning systems.

2. In settings where merging is insufficient, we show that existing localization methods are a promising solution, but may still perform far worse than the local models. We propose optimizing masks on local data, which fully recovers or even exceeds local performance.

3. Since localization introduces new costs associated with local masks, we propose clustering tasks and learning a mask for each cluster rather than task. We find that clustering tasks with similar features or task vectors can significantly improve utility over random clustering.

4. Finally, we evaluate the utility and costs of combining clustering and masking. While these two methods generally sacrifice some utility compared to storing all of the local models, they overall improve upon the efficiency-utility tradeoff offered by existing baselines.

## 2 RELATED WORK

**Machine Unlearning.** Unlearning methods can be broadly categorized as exact or approximate. Standard approaches for *exact* machine unlearning tend to have high computational costs from re-training over a large retain set or high storage costs from training ensembles on disjoint shards of data (Bourtoule et al., 2021; Yan et al., 2022; Chen et al., 2022; Li et al., 2024; Chowdhury et al., 2024). On the other hand, *approximate* unlearning methods do not provably remove the influence of data points from the model and are evaluated through empirical testing (Eldan & Russinovich, 2024; Liu et al., 2024; Maini et al., 2024). However, many prior works show that such approaches are brittle and can be easily attacked (Marchant et al., 2022; Bertran et al., 2024; Hu et al., 2024a;b). Unlearning is also a natural problem in distributed settings where users benefit from sharing their data; methods tailored to these settings can also be categorized as exact (Qiu et al., 2023; Xiong et al., 2023; Xia et al., 2024) or approximate (Wu et al., 2022; Halimi et al., 2022).

**Model Merging.** Early work in model merging averages the parameters of multiple models trained with different hyperparameters on the same data to improve generalization (Wortsman et al., 2022). Concurrent works extend merging multi-task learning by considering models trained on diverse tasks and merging them via a weighted or rescaled average (Matena & Raffel, 2022; Dimitriadis et al., 2023; Ilharco et al., 2023). Since then, many methods have been proposed to improve the quality of model merging, such as linearized finetuning (Ortiz-Jimenez et al., 2024), dropping parameters with sign conflicts (Yadav et al., 2024a), and sparsifying task vectors (Marczak et al., 2024; Yu et al., 2024; He et al., 2024; Davari & Belilovsky, 2025). Scaling merging to a large number of models is an open question; current works focus more on the benefits of scaling the model size while merging relatively few tasks (Yadav et al., 2024b). Besides a few works that need all the task vectors to selectively merge subsets/permutations of weights (Ainsworth et al., 2022; Stoica et al., 2023; Ye et al., 2023; Xu et al., 2024), the methods listed above are all directly applicable to unlearning. However, they all face the same limitation as simple averaging: performance degrades with scale.

**Model Localization.** Localization applies task-specific masks which can recover much of the performance lost during model merging (Wang et al., 2024; Huang et al., 2024). A few recent works utilize merging and localization in ways which are applicable for unlearning, but do not consider

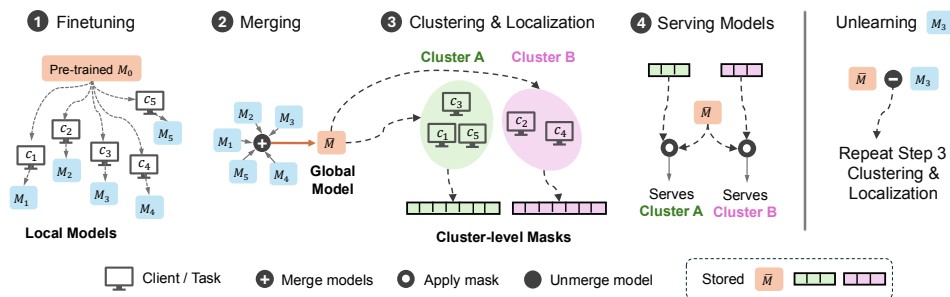

Figure 1: A step-by-step overview of ClAMU. Given multiple tasks which may need to be unlearned in the future, we (1) finetune a model separately on each task, (2) merge these models into one global model, (3) cluster similar tasks together, and (4) serve a uniquely masked model to each cluster. To unlearn a given task, we retrain its local model and unmerge (subtract) it from the global model.

unlearning and/or large-scale merging. Hu et al. (2024c); Lu et al. (2024) use input-based routing to select merging candidates, but consider only up to 4 and 8 tasks respectively. Zhang et al. (2024) uses extra storage to preserve specialization of individual experts, but only merges up to 4 experts.

## 3 ClAMU: Clustering, Averaging, and Masking for Unlearning

In this work, we focus on *exact* unlearning methods which have the benefit of provable unlearning by design. Given a model trained on the union of a retain set and forget set (data to be unlearned), exact unlearning produces a model which is identical to a model trained on only the retain set.

**Model merging is well-suited for exact unlearning.** As Figure 1 shows, merging is a framework which finetunes a pretrained model $M_0$ separately on several tasks, constructs residual *task vectors* $\tau_c = M_0 - M_c$, and then combines these to produce a multi-task vector $\overline{\tau} = \sum_{c \in [T]} \tau_c$ and a global model $\overline{M} = M_0 + \alpha\overline{\tau}$, where $\alpha$ is a hyperparameter (Ilharco et al., 2023). When our dataset is the union of several non-overlapping tasks $c_1, ..., c_T$ (e.g. data sources or clients), unlearning task $c_u$ updates the multi-task vector $\overline{\tau} - \tau_u = \sum_{c \in [T] \setminus \{c_u\}} \tau_c$ and the global model to match that from merging as if $c_u$ were never present. Furthermore, we can efficiently store $\overline{\tau}$, as it is a single parameter vector, as well as efficiently unmerge $M_u$ from $\overline{M}$ after retraining only on $c_u$.

**Localization recovers utility lost from merging.** When $\overline{M}$ performs poorly, localization methods additionally learn a mask $m_t$ for each task which approximates the local model weights once applied to the multi-task vector, i.e. $M_t \approx M_0 + m_t \odot \overline{\tau}$. TALL-masks constructs the mask $m_t = \mathbb{1}\{|\tau_t| \geq |\overline{\tau} - \tau_t| \cdot \lambda_t\}$ using a similarity threshold hyperparameter $\lambda_t$ (Wang et al., 2024), while EMR-merging simply uses sign agreement: $m_t = \mathbb{1}\{\tau_t \odot \overline{\tau} > 0\}$ (Huang et al., 2024).

Instead of optimizing for the similarity between $m_t \odot \tau_{\text{MTL}}$ and $\tau_t$, ClAMU directly optimizes the mask $m_t$ to minimize training loss on task $t$. To solve this high-dimensional discrete optimization problem, we learn a score vector $s_t$ in a similar fashion to prior work in neural network pruning (Ramanujan et al., 2020). Details on mask optimization are provided in Algorithm 2 of the appendix.

**Unfortunately, localization introduces additional costs.** Localization requires us to: (1) relearn the local masks after unlearning and (2) store masks during model serving. (1) can be done much more quickly than retraining a single model from scratch, as we can relearn the masks in parallel across all tasks. Meanwhile, the cost of (2) scales linearly with the number of tasks to be merged.

**Clustering improves utility and storage.** To reduce storage costs, we cluster tasks and learn *cluster-level* (rather than task-level) masks. We cluster tasks based on their hidden features or their task vectors. Compared to task-level masking, this not only gives model providers a flexible tradeoff between storage and utility, but can also improve utility by serving a common model to similar tasks.

### 3.1 When is model merging scalable?

We claim that the key factor affecting merging quality is data heterogeneity i.e. different local data generating process at each task. Merging degrades when data is highly heterogeneous i.e. "too different", an issue which is exacerbated as more tasks are merged. To illustrate this, Figure 2 tests 4

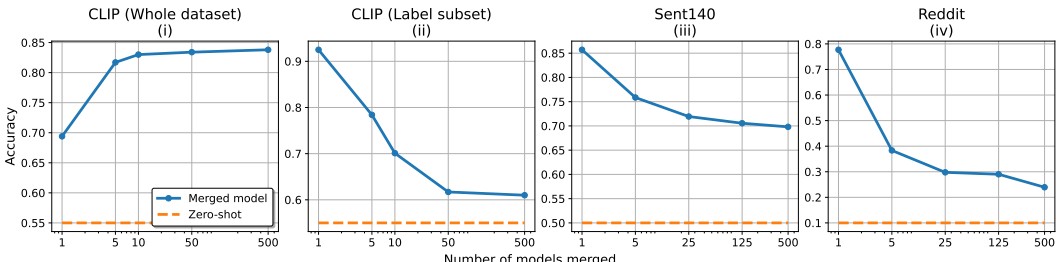

Figure 2: We evaluate accuracy (y-axis) after merging up to 500 models (x-axis, log-scale) trained on (i) task-level subsampled CLIP data, (ii) task-and-label-level subsampled CLIP data, (iii) Sent140, and (iv) Reddit. In settings (ii) and (iv), merging quickly degrades to zero-shot accuracy, while in setting (i) merging improves over individual models and in setting (iii) stabilizes at non-trivial utility.

distinct settings (i-iv). For the first two settings, we split three vision datasets (DTD, RESISC45, and MNIST) into 500 tasks by assigning each task to one of these three datasets and then partitioning each dataset such that each task gets 100 unique training and 10 validation examples. In setting (i), each task samples data uniformly at random from the dataset it is assigned to. In setting (ii), we use a finer-grained partitioning where each dataset is split into disjoint label subsets and each task samples from one of these label subsets. In setting (i), local performance starts out poor and improves as more models are merged. The opposite is true in setting (ii); local performance is high, but this drops sharply after merging. For the latter two setings, we use Sent140 and Reddit, which are text datasets naturally generated by users on social media sites. In Sent140 (iii), the labels are balanced across all tasks, while in Reddit (iv), each task samples its data from a subset of two labels. In a similar observation to settings (i,ii), we find that merging is more harmful when individual tasks' data is restricted to label subsets within a dataset. Full dataset details are in Table 6 of the appendix.

## 4 RESULTS

In this section, we present experiments on our two major methods and how they improve over baselines. Data-driven clustering outperforms random clusters or even the local models (i.e. no clustering), while optimizing local masks outperforms heuristic localization methods. `ClAMU` combines these two methods to achieve large improvements in utility. In Figure 3, we compare `ClAMU` to other baselines in the CLIP label subset (ii) setting. A single global model has the least storage but low utility, local models have high utility but are prohibitive to store, and TALL/EMR strike a balance between these two extremes. Meanwhile, `ClAMU` simultaneously outperforms local models while also costing less storage than other localization baselines. Additionally, `ClAMU` offers a flexible tradeoff of these two constraints for practitioners to choose from.

### 4.1 CLUSTERING TRADES STORAGE FOR UTILITY

Because the performance of the merged model can be much worse than applying each local model to its respective task, we consider a flexible tradeoff between these two extremes by storing multiple merged models. More specifically, we cluster tasks and then merge the task-level models within each cluster. We run experiments on Reddit where we train a GPT2-Small model individually on 500 tasks and then cluster these tasks into 4, 20, or 100 clusters. Finally, we merge the models within each clus-

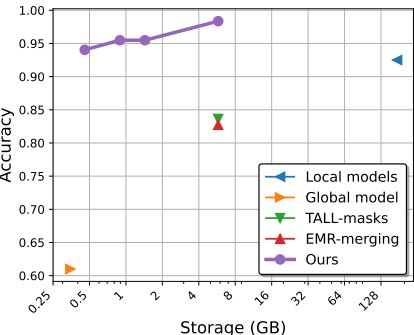

Figure 3: `ClAMU` combines clustering and localization to achieve strong performance while only storing a global merged model and cluster-level masks.

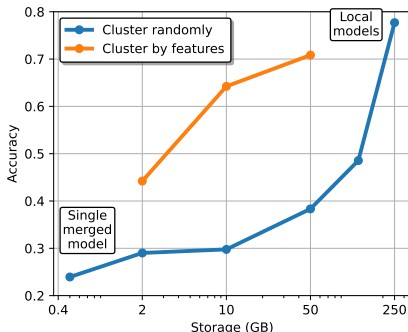

Figure 4: We evaluate the storage vs. accuracy of clustering tasks and storing a merged model for each cluster.

ter and evaluate each merged model on the tasks within its corresponding cluster. We compare the utility of these models after running the experiment with two clustering strategies: random versus feature-based clusters. When clustering by features, we use the hidden features at the last layer and last token of the model. Figure 4 shows that feature-based clustering is extremely useful and offers a 12-30% increase in accuracy over random clustering, depending on the number of clusters.

## 4.2 OPTIMAL LOCALIZATION

Localization can be viewed as either reducing the storage cost of local models (replacing them with local masks) or recovering performance of the globally merged model (at an extra storage cost). To demonstrate the effectiveness of localization, we evaluate sequence generation on TOFU in Table 1. Each local model can be trained to 100% local probability, but this drops sharply after merging. By using a larger model and optimizing the mask, we can recover 100% probability.

| TOFU | GPT2-Small (125M) | | | | GPT2-XL (1.5B) | | | |
|---|---|---|---|---|---|---|---|---|
| #Tasks | Global | TALL | EMR | ClAMU | Merged | TALL | EMR | ClAMU |
| 2 | 0.71 | 0.71 | 0.80 | **1.00** | 0.92 | 0.92 | 0.96 | **1.00** |
| 5 | 0.20 | 0.20 | 0.40 | **0.94** | 0.47 | 0.98 | 0.76 | **1.00** |
| 50 | 0.06 | 0.11 | 0.18 | **0.40** | 0.13 | 0.54 | 0.38 | **1.00** |
| 200 | 0.06 | 0.11 | 0.17 | **0.35** | 0.06 | 0.53 | 0.33 | **1.00** |

Table 1: We evaluate *answer generation probability* on TOFU. Each author's local model can be trained to 1.00 probability. The global model, TALL, and EMR all have limited utility as the number of clients increases, while ClAMU achieves 100% probability.

## 4.3 ClAMU: CLUSTERING AND MASKING

Motivated by the strong individual results of clustering and localization, we combine these two methods and learn *cluster-level masks*. In this experiment, we use the same setup and feature-based clusters as in Figure 4. We merge all 500 models into a single global model and then use either ClAMU, TALL, or EMR to learn a mask for each cluster. To learn the mask for TALL and EMR, we treat the per-cluster merged models from Figure 4 as the local task vector. Figure 5 shows the accuracy of each of these methods while varying the number of clusters from 4 to 100. While storing all 500 masks is already quite efficient ($x = 8$ GB), clustering can reduce this even further e.g. to 2 GB. Depending on the number of clusters, ClAMU improves accuracy by 5% to 20% over the two baselines. Because all three methods share the same clustering, these gains are entirely due to optimizing the mask.

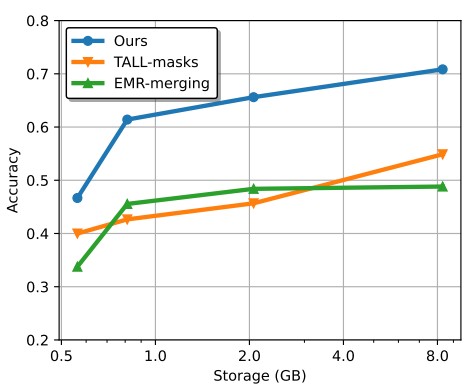

Figure 5: We train a model for each task on Reddit, merge these 500 models, then cluster the tasks based on their training features. Finally, we learn a mask for each cluster using either ClAMU, TALL, or EMR. While TALL and EMR are designed for task-level masking, we merge models within each cluster to provide these methods with a cluster-level model for localization.

## 4.4 UNLEARNING COSTS

Finally, we evaluate the unlearning costs and post-unlearning performance of merging alone on Sent140 and task-level CLIP. Since we observe in Figure 2 that merging more models hurts in several settings, the opposite is expected as we perform unlearning in these settings. In settings besides CLIP (i), the quality of the merged model will actually increase as we unlearn, because there is less interference between the remaining models that participate in merging.

For unlearning costs, we report the accumulated "Gradient Steps" required to retrain unlearned models in Table 2. This is simply the number of unlearned tasks × the number of steps used to train a local model (20 steps for Sent140 and Reddit, 500 for CLIP). Because merging only stores a single model, the only baseline with equal storage is training a single model from scratch. However, this cost would be orders of magnitude greater than merging-based unlearning. For example, unlearning 250 tasks (retraining from scratch after each task is unlearned) would require sequential training over $\sum_{i=250}^{499} i = 93625$ tasks worth of data, while merging-based unlearning would only need to train over the 250 tasks to be unlearned, a **374.5×** difference in computational cost.

| Dataset | #Tasks Unlearned
#Tasks Remaining | 0
500 | 250
250 | 375
125 | 475
25 | 495
5 | 499
1 |
|---|---|---|---|---|---|---|---|
| Sent140 | Accuracy
Gradient Steps | 69.8
0 | 70.1
5K | 70.8
7.5K | 71.9
9.5K | 75.9
9.9K | 85.7
10K |
| CLIP (i) | Accuracy
Gradient Steps | 83.8
0 | 83.8
125K | 83.9
188K | 83.2
238K | 81.7
248K | 69.4
250K |

Table 2: We evaluate the post-unlearning utility and cost of unlearning when only performing model merging. "Accuracy" is the average accuracy across all remaining tasks. "Gradient Steps" is the number of steps needed to retrain the local models and perform unlearning.

Next, we evaluate the costs of unlearning when applying both merging and localization. Although we must relearn the local model (TALL) or local mask (ClAMU) for each task after unlearning, this relearning can be done in parallel across each task's data. Therefore, we report the number of "Parallel Steps" required for unlearning, which is 2× that of the merging-only approach. We require 1× steps to unlearn a given model, and then another 1× steps to relearn the masks in parallel. We use an equal number of steps to retrain the local model for TALL and the mask for ClAMU .

| Method | #Tasks Unlearned
#Tasks Remaining | 0
500 | 250
250 | 375
125 | 475
25 | 495
5 | 499
1 |
|---|---|---|---|---|---|---|---|
| | Parallel Steps | 0 | 10K | 15K | 19K | 19.8K | 20K |
| No clusters | | | | | | | |
| TALL
ClAMU | Storage (GB)
Accuracy | 100
64.0
80.0 | 53
63.4
79.4 | 29.4
65.3
79.9 | 10.7
65.2
78.6 | 6.9
73.0
82.0 | 6
82.6
82.6 |
| 20 clusters | | | | | | | |
| ClAMU | Storage (GB)
Accuracy | 9.8
70.3 | 9.8
71.2 | 9.8
72.2 | 9.8
77.8 | 6.9
82.0 | 6
82.6 |

Table 3: We evaluate the post-unlearning utility and cost of unlearning when performing both model merging and localization on Reddit with a GPT2-XL model.

## 5 CONCLUSION AND FUTURE WORK

In this work, we propose using model merging and localization for exact unlearning. While merging is a very practical framework for exact unlearning, we show that the merged model suffers when too many heterogeneous data sources are merged. Based on this observation, we adapt merging and localization for the purposes of large-scale unlearning. We propose ClAMU, a method that improves (1) merging quality via clustering and (2) localization quality via direct optimization of mask variables, and we validate the effectiveness of ClAMU across both vision and language tasks. Overall, our work makes an important first step in identifying both the strengths and limitations of merging for exact unlearning, and proposes modifications that allow merging to achieve strong utility at scale. Finally, our results suggest that to make model merging truly effective for unlearning, future work should focus on (1) ways to improve the quality of the merged model and (2) modifications for recovering utility which are amenable to efficient unlearning.

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

# A APPENDIX

**Algorithm details.** One key detail of `ClAMU` is that finetuning must be fully deterministic. In order to guarantee that a model is properly unlearned, we must obtain the original weights that were used during merging and subtract them from the model.

We also provide more details on mask optimization: when computing the mask (L20), $\sigma$ is the sigmoid function. To compute the score update (L24), we backpropagate the loss $\mathcal{L}$ to the parameters of $M_k$ to obtain a gradient $\frac{\partial \mathcal{L}}{\partial M_t}$. In order to backpropagate through the indicator function, we substitute its gradient with 1 (i.e. the straight-through estimator). Backpropagating to the scores $s_k$ yields the update rule $s_k \leftarrow s_k - \eta(\frac{\partial \mathcal{L}}{\partial M_k}) \odot (\sigma(s_k) \odot (1 - \sigma(s_k))) \odot \overline{\tau}$, where $\eta$ is the learning rate.

---

**Algorithm 1:** PyTorch-like pseudocode helper functions for `ClAMU`

1 Require: $T$ (task indices), $\{c_t\}_{t \in T}$ (tasks), $M_0$ (pretrained model), $E_{\text{tune}}$ (finetuning steps), $K$ (num. of clusters), $E_{\text{mask}}$ (mask optimization steps)
2 **Function** `Finetune`$(t)$:
3     $M_t \leftarrow$ Finetune $M_0$ on $c_t$ for $E_{\text{tune}}$ steps (fully deterministic)
4     $\tau_t \leftarrow M_t - M_0$
5     **return** $\tau_t$
6 **Function** `Merge`$(\{\tau_t\}_{t \in T})$:
7     $\overline{\tau} \leftarrow \sum_{t \in T} \tau_t$
8     **return** $\overline{\tau}$
9 **Function** `Cluster`$(\{\tau_t\}_{t \in T})$:
10     $A \leftarrow [0]_{T \times T}$
11     **for** $i=1..T$ **do**
12         **for** $j=i..T$ **do**
13             $A_{i,j}, A_{j,i} \leftarrow \tau_i \cdot \tau_j / (||\tau_i|| ||\tau_j||)$   # cosine similarity affinity matrix
14     $C_{1..K} \leftarrow$ `sklearn.cluster.SpectralClustering`$(A)$
15     **return** $C_{1..K}$
16 **Function** `Localize`$(\overline{\tau}, C_{1..K})$:
17     $s_k \leftarrow \vec{0}$
18     **for** $k = 1..K$ *in parallel* **do**
19         **for** $i = 1..E_{mask}$ **do**
20             $m_k \leftarrow \mathbb{1}\{\sigma(s_k) > 0.5\}$
21             $M_k = M_0 + m_k \odot \overline{\tau}$
22             $x, y \leftarrow$ sample batch of data from $C_k$
23             $\mathcal{L} \leftarrow$ `torch.nn.functional.cross_entropy`$(M_k(x), y)$
24             $s_k \leftarrow s_k - \eta(\frac{\partial \mathcal{L}}{\partial M_k}) \odot (\sigma(s_k) \odot (1 - \sigma(s_k))) \odot \overline{\tau}$
25     **return** $\{m_k\}_{k=1}^K$
26 **Function** `Unlearn`$(\overline{\tau}, t)$:
27     $\tau_t =$ `Finetune`$(t)$
28     $\overline{\tau} \leftarrow \tau - \tau_t$
29     $T \leftarrow T \setminus t$
30     $C_{1..K} \leftarrow$ `Cluster`$(\{\tau_t\}_{t \in T})$
31     $m_{1..K} \leftarrow$ `Localize`$(\overline{\tau}, C_{1..K})$
32     **return** $\overline{\tau}, m_{1..K}$

---

**Algorithm 2:** PyTorch-like pseudocode for `ClAMU` (pre-unlearning)

1 Require: $T$ (task indices), $\{c_t\}_{t \in T}$ (tasks), $M_0$ (pretrained model), $K$ (clusters)
2 **for** $t \in T$ **do**
3     $\tau_t \leftarrow$ `Finetune`$(t)$
4 $\overline{\tau} \leftarrow$ `Merge`$(\{\tau_t\}_{t \in T})$
5 $C_{1..K} \leftarrow$ `Cluster`$(\{\tau_t\}_{t \in T})$
6 $m_{1..K} \leftarrow$ `Localize`$(\overline{\tau}, C_{1..K})$
7 **return** $\overline{\tau}, m_{1..K}$

**Task format impacts merging and localization.** We find that on Sent140, it is better to learn a classification layer but the differences are relatively small. On Reddit, it is significantly better to finetune the language modeling head.

| Clustering | Random clusters | | Cluster by features | |
|---|---|---|---|---|
| Model | CLS Head | LM Head | CLS Head | LM Head |
| Local | 36.4 | **38.3** | 64.8 | **70.8** |
| TALL | 25.1 | 27.8 | 40.1 | 45.7 |
| EMR | 24.6 | 28.1 | 40.8 | 48.4 |
| ClAMU | 31.3 | **43.1** | 48.3 | **65.6** |

Table 4: On Reddit, we merge 500 fully finetuned GPT2-Small models and learn masks for 100 clusters. We compare two different methods of modeling the classification output during finetuning: We either finetune a (randomly initialized) classification head to directly output the class index or finetune the (pretrained) language modeling head to output the class name as text. Using the LM head results in better cluster-level merged models than the CLS head ("Local"), under both random and data-driven clusters. These differences become even more apparent when applying localization.

| Clustering | Random clusters | | Cluster by features | |
|---|---|---|---|---|
| Model | CLS Head | LM Head | CLS Head | LM Head |
| Local | **67.8** | 63.2 | **69.4** | 66.3 |
| TALL | 64.4 | 61.7 | 65.3 | 62.8 |
| EMR | 63.3 | 59.1 | 64.2 | 61.2 |
| ClAMU | **65.7** | 64.7 | **66.3** | 66.3 |

Table 5: We run the same experiment in Table 5 on Sent140. While the classification head does better after merging ("Local") and applying localization baselines (TALL and EMR), the two modeling choices perform similarly when masking with ClAMU.

**Effect of model size**. Models with more pretrained knowledge generally have better utility after merging. In Figure 7, we compare GPT2-Small (125M params) and GPT2-XL (1.5B params). Using a GPT2-XL model improves the utility of local fientuning ($x = 1$) by 5%, while the improvement for the merged model ($x = 500$) is slightly greater at 7%.

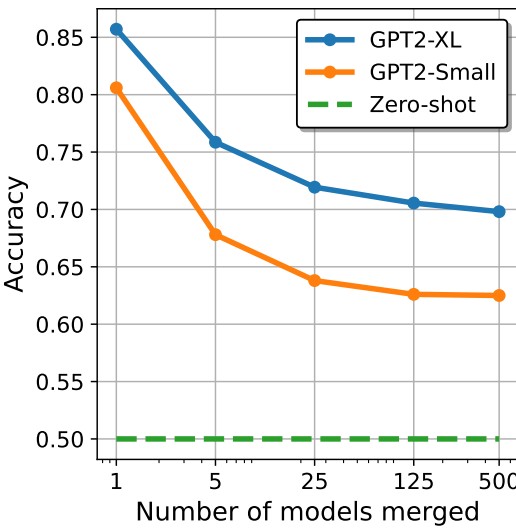

| Dataset | Type | Tasks | Labels (global) | Labels (per task) | | Task Size | |
|---|---|---|---|---|---|---|---|
| | | | | Min | Max | Min | Max |
| CLIP (i) | Image | 500 | 102 | 10 | 47 | 100 | 100 |
| CLIP (ii) | Image | 500 | 102 | 2 | 5 | 100 | 100 |
| TOFU | Text | 200 | 50256 | ? | ? | 20 | 20 |
| Sent140 | Text | 500 | 2 | 2 | 2 | ? | ? |
| Reddit | Text | 500 | 10 | 2 | 2 | 100 | 100 |

Table 6: TOFU is a sequence generation task, which requires exactly matching an output sequence of GPT2 tokens. On the image tasks, we train a ViT-B-32 (87M params) vision encoder with a classification head initialized using a CLIP text encoder. On the text tasks, we train GPT2-Small (125M params) and GPT2-XL (1.5B params) models.

