# OpenReview forum: "Exact Unlearning of Finetuning Data via Model Merging at Scale"
_ICLR.cc/2025/Workshop/MCDC — MCDC @ ICLR 2025_

### Official Review · Reviewer_t9VV · 2025-02-23

**Rating:** 4
**Confidence:** 3
**Fit:** 4

**Summary:**

The authors propose ClAMU for efficient and exact data unlearning for merged models. ClAMU proceeds in two steps, 1) tasks are clustered together and cluster-level masks are learnt, and then 2) masks are optimized on training data. The authors propose that this method improves overall utility of merged models, reduces storage requirements, and reduces unlearning costs.

**Reason For Giving A Higher Score:**

Unlearning is a timely and important topic and the focus on scalability and application of model merging is interesting.

**Reason For Giving A Lower Score:**

I think there are quite large issues regarding what the paper claims to propose (both model merging for unlearning and a novel masking method), the problems the method aims to solve (brittleness in unlearning methods), and how the authors use experiments to justify its claims. If I have misunderstood the logic of the paper, then I would appreciate clarification from the authors on the identified weaknesses and I would be happy to re-evaluate my score.

**Strengths And Weaknesses:**

Strengths:
1. The paper tackles an important topic area (unlearning), and the use of merging and focus on scalability are interesting and relevant.

2. Experimental validation of the benefits of their ClAMU over baselines with regard to accuracy and storage appears thorough.

Weaknesses:
1. The problem setting is not clearly defined. The authors note that a major limitation of unlearning methods is their brittleness such that models can easily be attacked to reveal the unlearnt information. But then the paper doesn't seem to address this identified shortcoming, instead focusing on storage, utility and cost.

2. Lack of clarity in presentation and unsupported claims. In the introduction, the authors claim to propose the technique of model merging for unlearning (line 37), however the experimental results centre on their masking approach, as opposed to justifying the more core concept of merging for unlearning. Furthermore, the authors state outright merging is well-suited to unlearning (line 132), but, again, this is a proposition that needs justification and not something that can just be asserted.

3. Missing experiments. I would've expected experiments demonstrating the efficacy of removing unwanted knowledge from the model using the proposed approach vs baseline unlearning methods. However, experiments seem to center on the improved accuracy-storage tradeoff of their masking method and then later unlearning cost.

(More minor point): line 48 notes that prior work has considered merging small numbers of models, but theres no citation provided.

**Suggestions:**

1. The authors should consider rewriting the motivation of their method, as the claimed brittleness of unlearning methods is not addressed

2. The authors should consider clarifying whether they are proposing the concept of model merging for unlearning, or whether they are taking merging for unlearning as a given and proposing a novel masking technique. If the former, then the authors should consider experimental justification for this claim, including experiments demonstrating the efficacy of removing knowledge by forms of merging. If the latter, then the authors need to rewrite the introduction to clarify that they do not propose merging for unlearning, and just propose a novel masking technique. This would also then require some discussion of merging for unlearning in the related work.

---

### Official Review · Reviewer_wc1z · 2025-02-25

**Rating:** 7
**Confidence:** 3
**Fit:** 4

**Summary:**

This paper tackles exact unlearning in the context of finetuned models. The key idea is to store a single global model obtained by merging many local finetuned models. Because each local model’s contribution is an additive task vector [1], unlearning that subset can be done exactly by subtracting its vector. While conceptually straightforward, merging can degrade performance across many heterogeneous tasks. The main contributions lie in introducing (1) Clustering tasks to reduce storage overhead and (2) Masking to recover local performance from a single merged checkpoint.


***
[1] Ilharco et al, “Editing Models with Task Arithmetic”, ICLR, 2023.

**Reason For Giving A Higher Score:**

Please refer to the strengths.

**Reason For Giving A Lower Score:**

Please refer to the weaknesses.

**Strengths And Weaknesses:**

**Strengths**
- The paper is well-written and easy to follow. Its motivation is also clear and straightforward.
- The paper suggests concise methods (e.g., clustering and masking) to address the scalability of merging-based unlearning approaches, which are sound and easy to understand.
- The paper shares a number of interesting, sound empirical results that support the authors’ claim.

***

**Weaknesses**
- The objective of the paper is puzzling. For instance, why is Sec 3.1 presented in the main paper? Is it closely related to the other parts of the paper?  Furthermore, the paper opens Sec. 3 by illustrating the issues associated with merging a large number of models (500) and suggests that existing approaches (e.g., localization) show some significant cost issues. If localization is indeed the cause of this issue, can we avoid using them? Refining the paper and clarifying the objectives would strengthen the paper substantially.
- The idea of unlearning using task arithmetic is not novel [1], and several concurrent works [2,3] are present. While the paper assumes a unique setting where a large number of models are merged, there is a lack of elaboration on why this setting is probable and important.

- The paper claims that it can reduce the storage cost of saving model weights by substituting them with optimizable masks. However, wouldn’t this create additional costs in optimization (training)? While I believe the optimization costs would not surpass the gains of using masks, I would like to see how much they cost. Furthermore, I would like to know the details of clustering high-dimensional task vectors (e.g., is every layer compared? How is the cost of clustering?)

- Lack of theoretical analysis. While post-training (especially task arithmetic) is a dominantly experimental field, I suggest the authors add a theoretical analysis. Again, consider this a minor weakness as the reviewer is aware of the experimental nature of the post-training literature.


***
[1] Ilharco et al, “Editing Models with Task Arithmetic”, ICLR, 2023.

[2] Kim et al, “NegMerge: Consensual Weight Negation for Strong Machine Unlearning”, ArXiv, 2024.

[3] Kadhe et al, “Split, Unlearn, Merge: Leveraging Data Attributes for More Effective Unlearning in LLMs”, ArXiv, 2024.

**Suggestions:**

The topic of exact unlearning is an important topic that requires significant attention, as it is directly related to illuminating how learned knowledge is stored in the model. While this paper shares very insightful empirical results, there is a lack of theoretical analysis on the topic. We strongly recommend the authors to include a theoretical analysis or at least a more detailed empirical analysis on the model's weight/parameter space.

---

### Official Review · Reviewer_uHZu · 2025-03-01

**Rating:** 7
**Confidence:** 3
**Fit:** 5

**Summary:**

This paper presents ClAMU, a novel framework for improving machine unlearning through model merging and localization. ClAMU addresses the challenge of efficiently removing data influence from fine-tuned models, especially when handling numerous tasks. At a high level, it employs clustering to group similar tasks and optimizes masks at the cluster level to improve utility and reduce storage costs. The framework's effectiveness is validated across both vision and language tasks, demonstrating improved accuracy and efficiency in unlearning. Experimental results show that ClAMU outperforms existing baselines in utility, storage, and unlearning cost. Additionally, the paper examines how data heterogeneity affects merging quality, suggesting future research directions for improving model merging in unlearning.

**Reason For Giving A Higher Score:**

Please see the strengths.

**Reason For Giving A Lower Score:**

Please see the weaknesses.

**Strengths And Weaknesses:**

Strengths:

- The paper addresses the challenge of efficiently updating a fine-tuned model to eliminate the influence of specific training data. This is crucial for complying with privacy regulations and reducing risks associated with fine-tuning.
- The paper introduces ClAMU, a novel framework for improving machine unlearning by integrating model merging and localization techniques. It introduces two key innovations: (1) task clustering, where similar tasks are grouped into clusters, and masks are learned at the cluster level instead of the task level. (2) improved localization, which optimizes masks directly on the training data for improved performance. These innovations allow ClAMU to outperform existing baselines in utility, storage, and unlearning cost.
- The paper investigates model merging with a large number of models (up to 500) and examines when merging can effectively achieve exact unlearning. It finds that the success of merging depends on data heterogeneity—performance remains high when data is relatively homogeneous but degrades significantly when data varies widely across tasks.
- The paper evaluates the combination of clustering and masking, demonstrating an improved efficiency-utility tradeoff compared to existing baselines.

Weaknesses:

- The paper demonstrates that merging quality varies significantly across tasks, with a noticeable degradation when the data is highly heterogeneous. However, the paper lacks a thorough analysis of how varying degrees of heterogeneity impact ClAMU's performance and how ClAMU handles extreme cases of data heterogeneity.
- ClAMU addresses localization costs through clustering tasks and learning cluster-level masks rather than task-level masks. However,  localization still introduces costs since local masks need to be relearned after unlearning and the storage cost scales with the number of clusters.
- The combination of clustering and masking generally sacrifices some utility compared to storing all local models.
- It is unclear how ClAMU handles scenarios where unlearning involves multiple tasks, particularly when it is difficult to determine which specific tasks to unlearn or the extent to which each task should be unlearned.

**Suggestions:**

- The authors should provide a more in-depth analysis of ClAMU's limitations and potential failure cases when handling highly heterogeneous data. Specifically, they should examine how data heterogeneity affects performance and offer clear guidelines for practitioners on managing such challenges effectively.
- The authors should clarify how ClAMU reduces costs compared to other localization methods.
- The authors should discuss the conditions under which sacrificing utility is acceptable and provide guidance on how practitioners can balance utility and storage trade-offs.
- It would be beneficial to explore how ClAMU handles unlearning in scenarios involving multiple tasks, as mentioned earlier.

---

### Decision · Program_Chairs · 2025-03-06

**Decision:**

Accept

**Comment:**

This paper tackles exact unlearning in the context of finetuned models by leveraging model merging and localization. Most of the reviewers liked the paper, found it relevant to the workshop, and recommended acceptance. We suggest the authors to incorporate the comments on reframing the paper brought up by the reviewer t9VV to further strengthen the paper. Overall, we're recommend to accept this work to the workshop.